# Amyotrophic Lateral Sclerosis and Autophagy: Dysfunction and Therapeutic Targeting

**DOI:** 10.3390/cells9112413

**Published:** 2020-11-04

**Authors:** Azin Amin, Nirma D. Perera, Philip M. Beart, Bradley J. Turner, Fazel Shabanpoor

**Affiliations:** Florey Institute of Neuroscience and Mental Health, University of Melbourne, Melbourne, VIC 3052, Australia; azina@student.unimelb.edu.au (A.A.); nirma.perera@florey.edu.au (N.D.P.); phil.beart@florey.edu.au (P.M.B.); bradley.turner@florey.edu.au (B.J.T.)

**Keywords:** motor neuron disease, amyotrophic lateral sclerosis, autophagy, therapeutics

## Abstract

Over the past 20 years, there has been a drastically increased understanding of the genetic basis of Amyotrophic Lateral Sclerosis. Despite the identification of more than 40 different ALS-causing mutations, the accumulation of neurotoxic misfolded proteins, inclusions, and aggregates within motor neurons is the main pathological hallmark in all cases of ALS. These protein aggregates are proposed to disrupt cellular processes and ultimately result in neurodegeneration. One of the main reasons implicated in the accumulation of protein aggregates may be defective autophagy, a highly conserved intracellular “clearance” system delivering misfolded proteins, aggregates, and damaged organelles to lysosomes for degradation. Autophagy is one of the primary stress response mechanisms activated in highly sensitive and specialised neurons following insult to ensure their survival. The upregulation of autophagy through pharmacological autophagy-inducing agents has largely been shown to reduce intracellular protein aggregate levels and disease phenotypes in different in vitro and in vivo models of neurodegenerative diseases. In this review, we explore the intriguing interface between ALS and autophagy, provide a most comprehensive summary of autophagy-targeted drugs that have been examined or are being developed as potential treatments for ALS to date, and discuss potential therapeutic strategies for targeting autophagy in ALS.

## 1. Introduction

### 1.1. Amyotrophic Lateral Sclerosis

Motor neuron (MN) diseases are a group of neurodegenerative diseases where MNs selectively degenerate. Amyotrophic Lateral Sclerosis (ALS, known as motor neuron disease in United Kingdom and Australia, and Lou Gehrig’s disease in the USA) is the most common form [1,2], with approximately 1–2 newly diagnosed cases in every 100,000 people internationally every year [3].

In ALS, both the upper MNs in the motor cortex and the associated corticospinal tract, and the lower MNs in the brainstem and spinal cord selectively degenerate. As a result, neuromuscular function deteriorates, evoking weakness, muscle wasting, and paralysis [2,4,5,6]. In the majority of cases, the disease manifests itself between the ages of 50 and 60 [5]. Typically, 3–4 years after symptom onset, respiratory muscles also atrophy, culminating in death [2,4,5,6]. However, in approximately one-third of patients where symptoms begin at the bulbar level with dysarthria and dysphagia, survival is shortened to an average of 2 years [7,8,9].

ALS is a complex and heterogeneous disorder with likely multiple causes. More than 90% of cases are sporadic (SALS) with no obvious family history of disease, and 10% of cases are familial (FALS) with one or more identifiable genetic mutations (Figure 1). The first discovered ALS-associated gene was *superoxide dismutase 1 (SOD1)* [10]. However, more recently, an intronic hexanucleotide (GGGGCC) repeat expansion in the non-coding segment of the *chromosome 9 opening reading frame 72 (C9orf72)* gene was identified as the most prevalent cause of FALS [11,12]. After *C9orf72* and *SOD1*, the two most common mutations implicated in FALS are in the genes *TAR DNA binding partner 43 (TARDBP)* [13] and *fused in sarcoma (FUS)* [14,15].

SALS and FALS are clinically indistinguishable, and the predominant cytoplasmic accumulation of ubiquitinated, hyaline, and skein-like aggregates within degenerating MNs and glial cells is a hallmark of both forms of ALS [17,18,19,20]. With the exception of SOD1- and FUS-linked ALS, the major pathological protein in all cases of ALS is TDP-43 and analysis of post-mortem tissues from ALS patients and mouse models has established that there is a direct correlation between MN loss and TDP-43 pathology [21,22].

Intracellular protein aggregates form when the level of misfolded proteins reaches a critical concentration, subsequently assembling into small soluble oligomers. Eventually, with time and increasing concentration of proteins, oligomers convert into the more metabolically stable insoluble aggregates [23,24,25]. According to the “seeding-nucleation” model, oligomerisation is a slow process as it is thermodynamically unfavourable. However, once an oligomeric seed is formed, it grows rapidly into high-molecular-weight protein complexes [26,27]. Consequently, the process of protein misfolding might begin years before the appearance of protein aggregates [28,29].

The exact role of these different aggregates in cells remains unknown. They can be initially cytoprotective by sequestering harmful protein species and burying the hydrophobic core of misfolded proteins, as opposed to having them exposed to the hydrophilic environment. Animal studies have supported this hypothesis, showing aggregates to be less toxic to cells than diffuse misfolded proteins [25]. Nonetheless, large intracellular aggregates disturb protein homeostasis, trigger cellular stress, and are closely associated with cell degeneration [17,24,25,30,31,32].

### 1.2. Cytoplasmic Protein Homeostasis and Degradation Pathways

Under normal physiological conditions, the protein quality control system circumvents aggregate-mediated toxicity by re-folding misfolded proteins using molecular chaperones or targeting them to one of two degradation pathways, the ubiquitin–proteasome system (UPS) or autophagy–lysosomal system [33].

The UPS degrades misfolded and short-lived soluble proteins through the protease complex [34]. The substrates have to be small enough to pass through the narrow pore of the proteasomal barrel. Studies in mouse models [35,36,37,38] and patients [39] indicate that the proteasome function is impaired in ALS spinal cord MNs.

In contrast to the UPS, autophagy does not have a size limitation for substrate clearance and can process large protein aggregates, not to mention organelles and intracellular bacteria [40,41,42,43]. The remainder of this review focuses on autophagy, its role in the pathogenesis of ALS, and its modulation as a possible therapeutic approach for the treatment of ALS.

There are three major sub-types of autophagy, namely, microautophagy, chaperone-mediated autophagy (CMA), and macroautophagy [41]. The major degradative autophagy pathway is macroautophagy (hereafter referred to as autophagy).

Autophagy is an evolutionary conserved intracellular process that purges cells of insoluble proteins, entire organelles that are damaged or superfluous such as mitochondria via mitophagy, or endoplasmic reticulum via reticulophagy, as well as toxic metabolites; cancerous cells; and intact invading microorganisms such as bacteria, viruses, and protozoa [44,45]. Basal autophagy plays a pivotal role in protein and organelle quality control [46], innate and adaptive immunity [47], metabolism [48], development, and differentiation [49,50,51,52]. As well as its housekeeping functions, autophagy is vital under conditions of stress or starvation [44,53].

Neuronal autophagy is particularly crucial as neurons are terminally differentiated and non-proliferating cells. Their non-dividing feature does not allow their intracellular contents such as protein aggregates to become diluted by cell division. Hence, in a study of autophagy-deficient mice, protein aggregates accumulated predominantly in neurons [54,55]. Moreover, the high energetic demands of neurons necessitate the efficient turnover of proteins and organelles for proper cell functioning [56,57,58].

The autophagy dynamic process, referred to as autophagy flux, comprises multiple steps (Figure 2). Autophagy-inducing signals are recognised by AMP-activated protein kinase (AMPK), the central regulator of cellular energy balance. Subsequently, the autophagy-activating kinase Unc-51 like autophagy activating kinase 1 (ULK1) is upregulated, and the autophagy-inhibitory kinase mammalian target of rapamycin complex 1 (mTORC1) is downregulated [59,60,61]. The ULK1 complex recruits phosphatidylinositol 3-kinase class 3 (PI3KC3) comprising vacuolar protein sorting 34 (Vps34), P150, and the core protein Beclin 1 to produce phosphatidylinositol 3-phosphates (PI3Ps) [62,63,64]. Thus, a cup-shaped pre-autophagosomal structure forms where essential autophagy-related proteins (Atgs) are recruited [65]. One such protein is C-terminus of microtubule-associated protein 1A/1B-light chain 3 (LC3), which is cleaved, activated, and lipidated to LC3-II and transferred to the phospholipid bilayer of the pre-autophagosomal structure [66]. For selective autophagy, membrane-bound LC3-II recruits autophagy receptors such as p62, which recognise and target specific cargo to the forming autophagosome [67,68,69]. Thereupon, LC3-II hemi-fuses membranes to expand and surround cargo, forming the double-membrane vesicle called the autophagosome (Figure 2) [70,71,72,73,74].

Autophagosomes mature and are trafficked to the lysosome-rich microtubule-organising centre (the perinuclear region) where they dock and fuse with lysosomes for the formation of autolysosomes [75,76,77,78,79,80,81]. Autolysosomal contents, including entrapped LC3 and p62, are degraded, and generated micromolecules such as amino acids and fatty acids are released back into the cytosol for reuse (Figure 2). As autophagy products become abundant, mTORC1 signalling switches on anabolic pathways for protein synthesis and cell growth [82,83,84].

### 1.3. Autophagy in ALS

Increasing evidence suggests that ALS is a disease of protein dyshomeostasis, with autophagy dysfunction playing an important role in the pathogenesis of ALS. Autophagy appears to be highly induced in ALS, similar to other neurodegenerative diseases such as Parkinson’s, Huntington’s, and Alzheimer’s diseases [85]. Autophagy proteins, particularly those that are involved in the early stages of autophagy, such as LC3, Beclin 1, p62, and Atg5–Atg12 complex, are elevated in spinal MNs of SALS and FALS patients [86,87] and animal models [86,87,88,89,90]. As the disease progresses to the late symptomatic stages, accumulation of these autophagic factors is also observed in glial cells such as astrocytes and microglia [88]. The upregulation of these proteins is not accounted for by an increase of their transcripts [88]. There is also evidence from studies using electron microscopy showing the build-up of autophagosomes, but not of their matured form, autolysosomes, in spinal MNs of late-symptomatic ALS animal models and autopsied patients [89,91]. In vivo autophagy imaging of the spinal cord of ALS mice has also shown an elevated green fluorescent protein (GFP)-LC3 signal in MNs, indicative of autophagosome accumulation, from early to late symptomatic stages of disease [88].

In contrast, autophagy degradation appears to be reduced in ALS, as there is a build-up of protein aggregates in MNs and surrounding glial cells in the brain and spinal cord of ALS animal models [92] and patients [17,18,19,20]. The accumulation of morphologically altered mitochondria [93,94,95,96,97,98,99] and dysfunctional endoplasmic reticulum (ER) [86,100,101,102,103,104], which are typically cleared through autophagy [105], further reinforces this view.

Autophagy represents a stress adaptation pathway. Therefore, the presence of stressors that are prominent in diseased cells such as damaged DNA, abnormal protein and organelle accumulation, and ER and oxidative stress can over-induce autophagy and autophagosome formation and impair the degradation process [106]. It has been postulated that excessive autophagosome formation/accumulation, exceeding their clearance rate, causes stress and leads to type II autophagic cell death [107] and apoptosis [108]. In fact, inhibiting autophagy induction has beneficial effects in some neurodegeneration disease conditions [109]. Similarly, suppressing autophagosome formation genetically or by administering n-butylidenephthalide at the pre-symptomatic stage of disease in ALS mice attenuated their pathology and extended their lifespan [110,111,112].

Multiple ALS-linked genes, such as *SQSTM1*, *OPTN*, and *TBK1*, encode for core autophagy proteins and others, such as *C9orf72*, *FUS*, *TDP-43*, *VAPB*, *UBQLN2*, *VCP*, *CHMP2B*, *ALS2*, *FIG4*, *TUBA4A*, *PFN1*, and *DCTN* have a functional role in autophagy. Therefore, mutations of these genes can cause impairment in different stages of the autophagy pathway (Table 1, Figure 2). Moreover, the misfolded or aggregated protein products of some ALS-causing genes that are not directly involved in the autophagy process, such as *SOD1*, can abnormally interact with autophagy proteins to dysregulate their activity [90,113,114,115,116,117,118,119,120,121] (Table 1).

Moreover, any of the 40 mutant genes that have been associated with ALS to date [16,122,123] (Figure 1) can potentially overwhelm the autophagy pathway, as mutations can decrease the stability of their protein products and increase their misfolding and aggregation propensity [124,125,126]. Compensatory mechanisms that upregulate the translation of the aggregate-entrapped proteins to prevent their loss-of-function further exacerbate aggregate formation as the prion-like characteristic of these proteins corrupts and converts the newly synthesised proteins into misfolded replicates [127,128,129,130,131,132,133,134,135,136].

Furthermore, akin to other neurodegenerative diseases, ALS is an age-related disease. During normal ageing in the human brain, autophagy genes such as Atg5, Atg7, and Beclin 1 are transcriptionally downregulated [137,138]. In addition, in aged animal models and human MNs, there is a build-up of intra-lysosomal aggregates in the form of lipofuscin granules, extra-lysosomal aggregates, and defective mitochondria, which can overwhelm and impede autophagy [139]. Therefore, the age-related decline of autophagy proteins and lysosomal proteolytic activity [140,141] and the upsurge of autophagy substrates could reduce autophagy activity and potentially explain the late manifestation of ALS.

### 1.4. Autophagy-Targeted Treatments for ALS

Thus far, no effective treatment exists for ALS, despite the discovery of this disease nearly two centuries ago. Designing treatments that target specific ALS-causing genes to reduce the protein aggregation load is a potentially valid therapeutic approach. However, multifactorial diseases such as ALS with various aetiologies, from genetic to environmental and age-related, require treatments that target pathogenic processes and their progression in a timely and appropriate manner. The disappointingly modest effect of riluzole [168] and edaravone, the only two approved treatments available for patients that act to rectify excitotoxicity and oxidative stress, respectively, implies that a more central pathway needs to be targeted [169,170].

Dysregulation in autophagy is emerging as a significant contributor of pathogenesis in diverse neuropathologies, including ALS. Therefore, interest continues to grow in autophagy as a likely beneficial therapeutic target [171]. Different autophagy-modulating agents have been tested in ALS patients and disease models, and their findings are summarised below (Figure 3, Table 2).

### 1.5. Caloric Restriction and Its Mimetics

Caloric restriction through dietary intake reduction, starvation, or physical exercise can successfully induce autophagy by reducing glucose and insulin levels and thus inactivating mTORC1 [215,216] (Figure 3). However, this method is not considered a useful therapeutic approach because it can have harmful effects if not tightly regulated. In particular, males can be vulnerable to nutritional and metabolic stress, as caloric restriction or exercise reduces the lifespan of male, but not female ALS mice (Table 2) [172,173,217]. Moreover, the caloric restriction approach is time-inefficient and it fails to significantly affect brain autophagy, conceivably because of the stable nutrient supply to the brain [218]. Nonetheless, caloric restriction mimetics such as rapamycin, trehalose, spermidine, resveratrol, and metformin have been used to induce the beneficial effects of caloric restriction without its adverse consequences (Figure 3, Table 2).

Rapamycin is the most well-known pharmacological inducer of autophagy. Its administration has shown to be protective in several neurodegenerative disease animal models [219,220,221,222]. However, the effect of rapamycin on ALS differs depending on the animal model tested. In mice overexpressing human wild-type TDP-43 in the forebrain, rapamycin rescued motor neuron function [176]. However, in mice expressing mutant SOD1, rapamycin had no effect [174,178] or had detrimental effects [177]. The negative impact of rapamycin treatment on mutant SOD1 mice was partly attributed to immunosuppression of neuroprotective regulatory T cells [178], suggesting that global mTOR inhibition may not be useful for ALS.

### 1.6. Hormone Therapy

Female sex hormones can increase mitochondrial efficiency; reduce oxidative stress; and, more importantly, increase autophagy, as opposed to androgens, which decrease autophagy [223]. Hence, progesterone and oestrogen modulators such as raloxifene and tamoxifen have been used to induce autophagy in ALS (Figure 3, Table 2). In fact, female contraceptives may have protective roles in ALS [224] and ablating oestrogen production in female mutant SOD1 mice accelerates their rate of disease progression [225,226] (Table 1).

Therefore, differences in gonadal hormones could explain the slightly higher prevalence of ALS in men than in women [227]. Evidence to support this notion includes the equality of ALS ratio between the two genders at menopause, possibly a result of the age-related decline in oestrogen and progesterone levels in women [190], as well as the more infrequent susceptibility of ALS in pre-menopausal women compared to postmenopausal women [189].

### 1.7. Anti-Cancer Therapy

Cancer treatment drugs such as bosutinib and dasatinib suppress the phosphorylation of two closely related non-receptor tyrosine kinases, Src and c-Abl, to increase autophagy levels (Figure 3, Table 2).

The involvement of Src and c-Abl in the regulation of cell proliferation, apoptosis, and angiogenesis make them fundamental for neuron development [191]. However, in mature and healthy neurons, their kinase activity tends to be inactive, whereas, in sporadic and familial cases of ALS (and patients with other neurodegenerative diseases), their level of activity is significantly elevated. This inappropriate activation has been associated with neuroinflammation; cell cycle arrest; apoptosis; and, more intriguingly, autophagy inhibition [191] (Figure 3).

### 1.8. Psychotropic Therapy

Several mood-stabilising drugs such as lithium, valproate/valproic acid, and carbamazepine (Figure 3, Table 2) deplete inositol triphosphate (IP3) and thus have been used to stimulate autophagy.

IP3 interacts with its receptor (IP3R) on the ER to sequester the autophagy protein Beclin 1 and abolish its autophagy-inducing properties. IP3R activation also releases calcium from ER stores, which promotes calcium efflux to organelles such as mitochondria, leading to an increase in ATP production and subsequently autophagy inactivation [228]. An increase in intracellular calcium can also activate calpains to cleave and inactivate autophagy proteins [229] (Figure 3).

It is interesting to note that the majority of the proposed autophagy activators have mood-stabilising or antidepressant effects, raising the possibility for the involvement of protein dyshomeostasis in the pathogenesis of affective disorders [230].

### 1.9. Anti-Hypertensive Therapy

Although the effect of calcium flux on autophagy is ambiguous [231], calcium channel antagonists such as verapamil, and intracellular calcium modulating agents rilmenidine and berberine, used for the treatment of hypertension, have been used to promote autophagy (Figure 3, Table 2).

As mentioned in the previous section, increased mitochondrial calcium levels have an inhibitory effect on autophagy. However, under stressed conditions such as in ALS animal models, where there is an accumulation of misfolded proteins, cytosolic calcium can also positively regulate autophagy [232].

### 1.10. Anti-Histamine Therapy

Anti-histamine medications such as latrepirdine and clemastine have been suggested as potential autophagy inducers (Figure 3, Table 2).

The histaminergic system is involved in a plethora of processes such as smooth muscle contraction, vasodilation, gastrointestinal tract, immunity, and circulation [233]. Recent evidence indicates that it also modulates autophagy through AMPK [234,235].

## 2. Considerations for Effective Targeting Autophagy in ALS

The above-mentioned studies collectively suggest that a number of critical factors need to be taken into account when considering autophagy treatments for ALS. Firstly, the stage of the disease when treatment commences and duration of treatment. This is because genetic inhibition of autophagy in motor neurons of mutant SOD1 mice revealed that autophagy can be protective early in disease progression, but detrimental late in disease [110]. It is noteworthy to mention that some drugs such as clemastine were beneficial with acute treatment, while long-term treatment to the late phases of disease has no effect on disease progression or survival [214]. The stage of disease is likely to reflect differences in pathological substrates accumulating in affected tissues, such as soluble and insoluble misfolded proteins, oligomers, protofibrils, fibrils, aggregates, and inclusions, which could affect autophagy capacity and efficiency in cells.

Secondly, the pathway that the treatment targets to stimulate autophagy needs to be considered. Drugs such as rapamycin, metformin, progesterone, tamoxifen, bosutinib, dasatinib, latrepiradine, and clemastine act through the mTORC1 pathway to induce autophagy. However, mTORC1 is a master regulator of many autophagy-independent pathways in the cell such as protein synthesis, immunosuppression, cell cycle, and many more [236]. For example, the beneficial effects of rapamycin might be negated by its immunosuppressive effects, which might explain why SOD1 mice lacking mature lymphocytes show improved survival, compared to SOD1 mice with their immune system intact [178]. Moreover, upon prolonged treatment with rapamycin, the second mTOR kinase complex (mTORC2) becomes inhibited. This action can cause a reduction in the cell survival signalling pathway, which accounts for the upregulated levels of apoptotic proteins observed in long-term studies with rapamycin [237,238]. Therefore, autophagy-independent effects of mTOR inhibition must be taken into account, or drugs that modulate autophagy independently of the mTOR pathway should be considered [239,240]. Examples of these treatments include trehalose, spermidine, resveratrol, raloxifene, lithium, valproate, pimozide, methotrimeprazine, fluphenazine, verapamil, and rilmenidine. Despite this, treatment with mTOR-independent autophagy inducers such as trehalose [179,180,181,182] or rilmenidine [210] have yielded contrasting outcomes in mutant SOD1 mice, highlighting that bypassing mTOR signalling to invoke autophagy may not be straightforward.

Thirdly, it is critical to consider the stage of the autophagy pathway that the drug is acting on. Treatments that upregulated the autophagic flux/degradation such as trehalose [179,180,181], spermidine [176], tamoxifen [176], progesterone [190], bosutinib [191], lithium [193,196], carbamazepine [176], fluphenazine [206], and clemestine [213,214] showed beneficial effects in ALS mice. In contrast, rapamycin [177,207] and pimozide [177,207] induced autophagy but suppressed autophagic degradation, evidenced by accumulation of autophagosomes and autophagy substrates such as p62 and protein aggregates. These treatments exacerbated disease in ALS animal models. It is also worth noting that autophagic substrates, critically misfolded proteins and aggregates, are not always measured in tissues of ALS mice treated with autophagy-modulating drugs, which is essential to correlate with reported neuroprotective and clinical effects.

Fourthly, results of treatments may differ depending on the experimental model employed. In the absence of a mutation, such as in FTLD-U/TDP-25 cell culture and animal models where there is accumulation of substrates, autophagy can be successfully upregulated, irrespective of where it is targeted [175,176,189,201,211]. However, in animal models driven by mutant transgenes, autophagy may be impaired at different stages, and thus responses to treatments will differ. For example, loss of Beclin 1 in SOD1^G93A^ and SOD1^G127X^ mice exacerbates disease [86], whereas it improves the phenotype of SOD1^G86R^ mice [86,121]. Similarly, the genotype of patients plays an important role in the outcome of clinical trials [241].

Lastly, these drugs are repurposed, and their original target might not be the brain (e.g., metformin). Therefore, they might not be able to sufficiently access the nervous system to affect neuronal autophagy [242]. For the treatment of ALS, it is crucial to be able to efficiently and adequately deliver therapeutics to the central nervous system, more specifically to MNs. This is one of the great challenges of treating neurological diseases; however, significant progress has been made in the development of various MN drug delivery systems [243,244,245].

Moreover, besides autophagy activation, some of these drugs target many other biological processes. Some of these off-target effects are desired and consistent with the multifunctional involvement of autophagy such as antioxidant, mitochondrial, and calcium effects (Figure 4), and some of which are harmful and unfavourable. For example, where researchers were vigilant enough to test the response of both male and female mice to a drug, it was shown that the effect of some treatments are sex-specific. Trehalose [180,181] and resveratrol [184,185,187], which can compete with oestrogen for binding with its receptor, and metformin [188] and latrepiradine [212], which inhibit oestrogen production, all demonstrated more protective effects in male mice compared to females.

## 3. Conclusions

In this review, we have summarised the main evidence for the enhancement of the autophagy pathway as a possible therapeutic strategy for the treatment of ALS. However, site of autophagy pathway dysfunction and the time at which autophagy becomes dysfunctional during the disease process needs to be elucidated before autophagy treatments can be used as a therapy for ALS. To date, there is no effective autophagy-inducing agent, and those currently being used are small molecules and drugs that often lack selectivity and possess undesirable side effects. Future studies should seek to develop more selective autophagy-enhancing drugs. Moreover, we speculate that autophagy is impaired, not at the autophagosome formation step, but at the autophagosome maturation/degradation step. Therefore, the late, degradative steps of autophagy may need to be targeted, rather than the early steps. In short, promotion of waste clearance and diminution of cytotoxicity is conferred by autophagosome accumulation.

Additionally, it is essential to note that currently there is no definitive diagnostic biomarker for ALS, and by the time the disease clinically manifests itself, more than 50% of neurons are lost [240]. However, the majority of the drugs discussed above were administered at pre-symptomatic or at the onset of symptomatic stage of the disease. To enhance the translational value of animal models, the potential drug needs to be administered to ALS mice at the post-symptomatic stage of the disease to better mimic the pathological conditions of when patients are likely to commence treatment. Meanwhile, reliable early diagnostic markers of disease onset need to be developed so that treatments can be initiated promptly. For example, a sensitive assay that can detect the presence of ALS-associated misfolded proteins early would allow their clearance before they accumulate into larger, degradation-resistant cargo, thus preventing the pathological cascade mediated by aggregate accumulation (Figure 4). Lastly, the effects of the drug should be studied in multiple models of ALS to account for heterogeneity of ALS and to better assess drug suitability. A successful autophagy-enhancing agent would have a potential therapeutic benefit to not only ALS patients but also patients with other proteinopathies that have protein aggregates as their cardinal feature.

## Figures and Tables

**Figure 1 cells-09-02413-f001:**
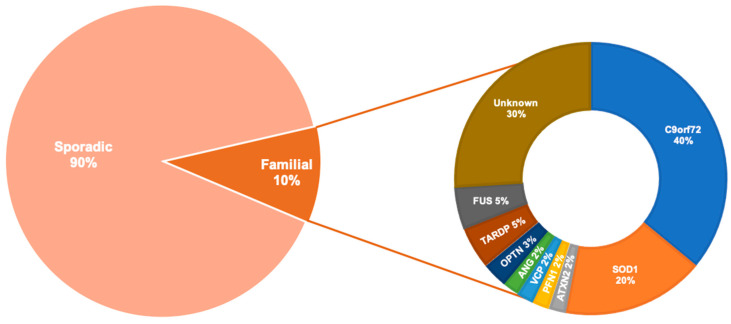
The prevalence of the most commonly known genetic causes of Amyotrophic Lateral Sclerosis (ALS). Other genes that are more rarely associated with ALS are not included in the diagram above are: sequestosome 1 (SQSTM1), dynactin subunit 1 (DCTN1), VAMP associated protein B and C (VAPB), D-amino acid oxidase (DAO), TATA-box binding protein associated factor 15 (TAF15), ubiquilin 2 (UBQLN2), heterogenous nuclear ribonucleoprotein A1 (hnRNPA1), heterogenous nuclear ribonucleoproteins A2/B1 (hnRNPA2B1), matrin 3 (MATR3), tubulin alpha 4a (TUBA4A), sec1 family domain containing 1 (SCFD1), myelin associated oligodendrocyte basic protein (MOBP), chromosome 21 open reading frame 2 (C21orf2), cyclin F (CCNF), NIMA related kinase 1 (NEK1), neurofilament heavy (NEFH), dnaJ heat shock protein family (DNAJ), EWS RNA binding protein 1 (EWSR1), senataxin (SETX), calcium- responsive transactivator (CREST), elongator acetyltransferase complex subunit 3 (ELP3), charged multivesicular body protein 2B (CHMP2B), alsin rho nucleotide exchange factor ALS2 (ALS2), sigma non-opioid intracellular receptor 1 (SIGMARI), FIG4 phosphoinositide 5-phosphatase (FIG4), spastic paraplegia 11 (SPG11), peripherin (PRPH), neuropathy target esterase (NTE), serum paraoxonase and arylesterase 1-3 (PON1-3), cholinergic receptor nicotinic alpha 3 (CHRNA3), cholinergic receptor nicotinic alpha 4 (CHRNA4), cholinergic receptor nicotinic beta 4 (CHRNB4), erb-b2 receptor tyrosine kinase 4 (ERBB4), coiled-coil-helix-coiled-coil-helix domain containing 10 (CHCHD10), amyotrophic lateral sclerosis 3 (ALS3), amyotrophic lateral sclerosis 7 (ALS7), amyotrophic lateral sclerosis 6-21 (ALS6-21), amyotrophic lateral sclerosis-frontotemporal dementia (ALS-FTD) [16].

**Figure 2 cells-09-02413-f002:**
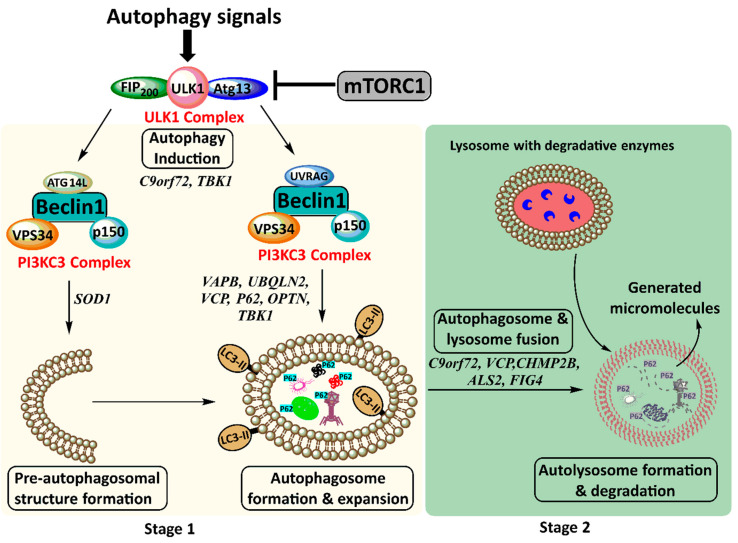
The two main stages of the autophagy pathway. Stage 1 involves autophagy induction where autophagy signals result in the formation of the Unc-51 like autophagy activating kinase 1 (ULK1) complex followed by the phosphatidylinositol 3-kinase class 3 (PI3KC3) complex. This leads to pre-autophagosomal structure formation and autophagosome formation and expansion. Stage 2 involves autophagosome transport for fusion with lysosomes. This leads to autolysosome formation and degradation of its contents. ALS-associated proteins that impair autophagy are indicated beside each step of the pathway. FIP200, FAK family kinase-interacting protein of 200 kDa; mTORC1, mammalian target of rapamycin complex I; ULK1, Unc-51 like autophagy activating kinase 1; p150/Vps15, vacuolar protein sorting 15; Vps34, vacuolar protein sorting 34; LC3-II, C-termini of microtubule-associated protein 1A/1B-light chain 3B; UVRAG, UV radiation resistance associated; Atg13, autophagy-related protein 13; Atg14L, autophagy-related protein 14.

**Figure 3 cells-09-02413-f003:**
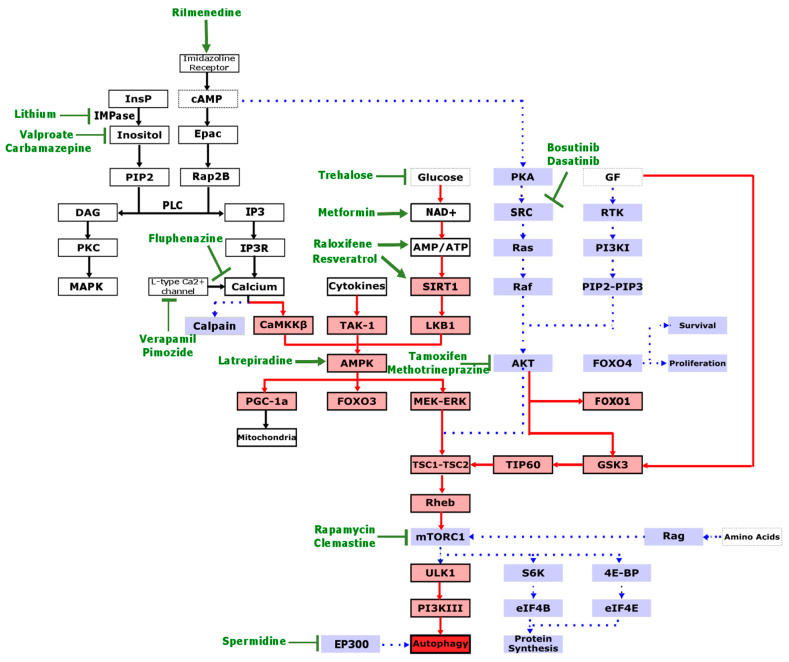
Autophagy-targeted treatments that have been studied in ALS. For autophagy to be activated, pathways depicted in red need to be upregulated and pathways depicted in blue need to be downregulated. The pathway that is targeted by the treatment is depicted in green (arrow if its effect is stimulatory and flathead if its effect is inhibitory). InsP, inositol phosphate; IMPase, inositol monophosphatase; PIP2, phosphatidylinositol 4,5- biphosphate; DAG, diacylglycerol; PKC, protein kinase C; MAPK, mitogen-activated protein kinase; cAMP, cyclic adenosine monophosphate; PLC, phospholipase C; IP3, inositol triphosphate; IP3R, inositol triphosphate receptor; CaMKKβ, calcium/calmodulin-dependent protein kinase 2; TAK-1, transforming growth factor beta-activated kinase 1; NAD+, nicotinamide adenine dinucleotide; SIRT1, NAD-dependent deacetylate sirtuin-1; LKB1, liver kinase B1; PKA, protein kinase A; AKT/PKB, protein kinase B; GF, growth factor; FOXO4, forkhead box 4, FOXO1, forkhead box 1; GF, growth factors; RTK, receptor tyrosine kinase; PI3KI, class I phosphoinositol 3-kinase; PIP3, phosphatidylinositol-3,4,5-triphosphate; GSK3, glycogen synthase kinase 3; TIP60, tat-interactive protein 60 kDa; TSC1–TSC2, tuberous sclerosis complex; Rheb, Ras homolog enriched in brain; mTORC1, mammalian target of rapamycin complex I; ULK1, Unc-51 like autophagy activating kinase 1; PI3KIII, class III phosphoinositol 3-kinase; S6K, ribosomal protein S6 kinase beta-1; 4E-BP, eukaryotic translation initiation factor 4E-binding protein 1; eIF4E, eukaryotic translation initiation factor 4E; EP300, E1A binding protein P300.

**Figure 4 cells-09-02413-f004:**
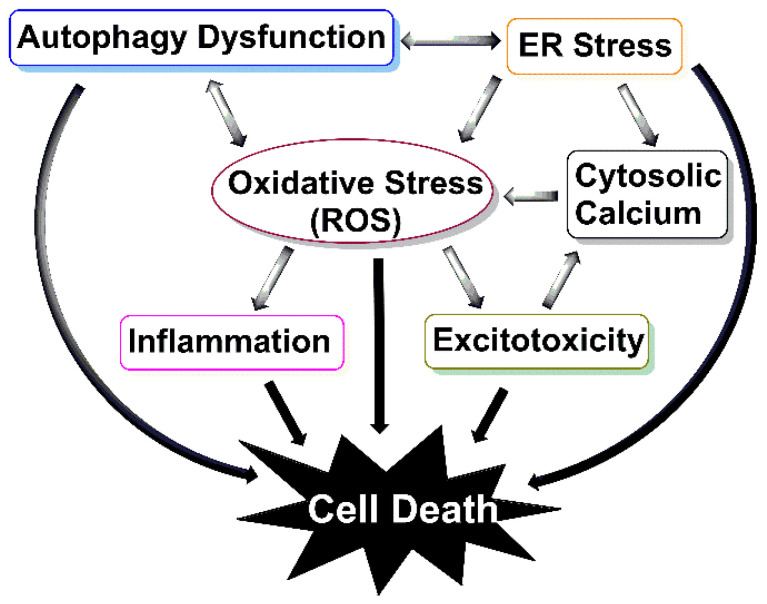
The interrelationship between autophagy and other pathogenic mechanisms in ALS. Dysfunction of autophagy, specifically, aggrephagy (aggregate-specific autophagy), mitophagy (aggregate-specific autophagy), and reticulophagy (ER-specific autophagy) leads to the build-up of protein aggregates, damaged mitochondria, and ER stress, respectively. Thus, inducing reactive oxygen species (ROS) production, which persistently builds up in ALS to inhibit autophagy, stabilises aggregates and induces inflammation and excitotoxicity. Excitotoxicity over-activates neurons, resulting in cytoplasmic and mitochondrial calcium augmentation and hence further mitochondrial damage. As a result of these impairments, cells undergo apoptosis and death.

**Table 1 cells-09-02413-t001:** The association between ALS genes and autophagy.

Gene	Protein	Function	Effect on Autophagy	Reference
*** SOD1 ***	*Superoxide dismutase 1*	Antioxidant enzyme: convert superoxide radical anions to oxygen/hydrogen peroxide.	Mutant associates with Beclin1–B cell lymphoma 2 (Bcl2) complex to disrupt the activity of Beclin1 → impairing autophagosome formation/expansion.	[121]
			Mutant and misfolded wild-type sequesters dynein proteins into inclusions → impairing retrograde transport and autolysosome formation.	[90,119,120]
			Mutant binds voltage-dependent anion-selective channel protein 1 (VDAC1) and Bcl2 → impairing mitophagy.	[114,115,116,117,118]
			Mutant binds to optineurin → impairing mitophagy.	[113]
*** C9ORF72 ***	*Chromosome 9 opening reading frame 72*	Guanine nucleotide exchange factor (GEFs) for Rab GTPases—allows trafficking via actin, regulates lysosome biogenesis and maturation.	Deficiency disrupts ULK1 complex trafficking → impairing autophagosome formation.	[120]
			Deficiency disrupts transfer of lysosomal proteins to lysosomes → impairing autolysosome formation.	[142]
			Deficiency disrupts its association with early endosomes for lysosome maturation → impairing autolysosome formation.	[142]
*** FUS ***	*Fused in sarcoma*	RNA-binding protein—regulates RNA splicing, processing, transport, and translation.	Mutant disrupts ER–Golgi transport and ER–mitochondrial interactions → impairing autophagosome formation.	[143,144]
			Deficiency decreases Parkin → impairing mitophagy.	[145,146]
*** TDP-43 ***	*TAR DNA binding partner 43*	RNA-binding protein—regulates RNA splicing, processing, transport, and translation.	Mutant disrupts ER–Golgi transport and ER–mitochondrial interactions → impairing autophagosome formation.	[143,144]
			Deficiency decreases Parkin → impairing mitophagy.	[145,146]
			Deficiency destabilises Atg7 mRNA and proteins → impairing autophagosome expansion.	[147]
			Mutant inhibits dynactin and HDAC6 → impairing retrograde transport for autolysosome formation.	[148,149]
*** SQSTM1 ***	*Sequestosome 1/p62*	Autophagy receptor—recruit cargo to autophagy.	Mutant disrupts recognition of ubiquitinated cargo and their delivery to autophagosomes → impairing cargo degradation.	[120]
***OPTN***	*Optineurin*	Autophagy receptor—recruits cargo to autophagy.	Mutant disrupts recognition of ubiquitinated cargo/mitochondria and their delivery to autophagosomes → impairing cargo degradation and mitophagy.	[120]
			Deficiency disrupts its interaction with myosin VI → impairing retrograde transport for autolysosome formation.	[120]
***TBK1***	*Tank-binding kinase 1*	Autophagy receptor activator—phosphorylates p62, OPTN, etc. for selective autophagy.	Mutant disrupts its kinase activity of autophagy receptors → impairing cargo/mitochondria degradation.	[150,151]
			Mutant disrupts its kinase activity of microtubule-binding proteins → impairing retrograde transport for autolysosome formation.	[120]
*** VAPB and VAPC ***	*VAMP-associated protein B and C*	ER proteins—activate endoplasmic reticulum-associated protein degradation (ERAD) and autophagy.	Mutant disrupts its interaction with PTPIP51 to allow ER–mitochondrial contact → impairing autophagosome formation.	[152,153]
*** UBQLN2 ***	*Ubiquilin*	ER protein—recognises ubiquitinated misfolded proteins.	Mutant disrupts recognition of ubiquitinated cargo and their delivery to autophagosomes → impairing cargo degradation.	[154,155]
***VCP***	*Valosin-containing protein*	ER protein—forms aggresomes and translocates them out of ER for degradation.	Mutant disrupts formation of aggresomes and translocates out of ER → impairing cargo degradation	[120,156]
			Mutant disrupts granulophagy (degradation of stress granules).	[157,158]
			Mutant disrupts its recruitment to damaged mitochondria → impairing mitophagy.	[159,160]
***CHMP2B***	*Charged multivesicular body protein 2 B*	Endosomal protein—generates multi-vesicular bodies (MVBs).	Mutant disrupts Rab5 conversion to Rab7 → impairing retrograde transport for autolysosome formation.	[161,162,163]
			Mutant disrupts dissociation of its ESCRT-III complex from the endosomal membrane MVB generation → impairing autophagosome maturation.	[161,162,163]
			Mutant disrupts biogenesis and maintenance of lysosomes → impairing autolysosome formation.	[161,162,163]
***ALS2***	*Alsin*	Endosomal protein—generates MVBs.	Mutant disrupts its GEF activity of Rab5 → impairing autophagosome maturation.	[164]
***FIG4***	*Polyphosphoinositide 5-phosphatase*	Endosomal protein—generates MVBs.	Mutant disrupts PI3P production → impairing autophagosome maturation.	[165]
***TUBA4A***	*Tubulin alpha 4A*	Cytoskeleton protein—allows transport within cells.	Mutant disrupts the dynamic and stability of microtubules → impairing retrograde transport for autolysosome formation.	[120,166,167]
***PFN1***	*Profilin1*	Cytoskeleton protein— allows transport within cells.	Mutant disrupts its interaction with actin → impairing retrograde transport for autolysosome formation.	[120,166,167]
***DCTN***	*Dynactin*	Cytoskeleton protein—allows transport within cells.	Mutant disrupts its complex formation with dynein to bind to microtubules → impairing retrograde transport for autolysosome formation.	[120,166,167]

**Table 2 cells-09-02413-t002:** The effect of autophagy-targeted treatments on ALS models. Sur., survival; Ref., reference; F, female mice; M, male; ↑, increased/restoration; ↓, decreased; ↔, unaffected; co, combination of treatments; *, p62 remained unchanged.

Treatment		Treatment Dosing	Model	Treatment Duration	Autophagy	MNs	Sur.	Ref.
**Caloric Restriction**			SOD1-G93A mice	Long-term (pre-symptomatic)		↑function	↓	[172]
			SOD1-G93A mice	Short-term (pre-symptomatic)			↓ (M)	[173]
			SOD1-H46R/H48Q mice	Long-term (pre-symptomatic)		↑function	↑	[174]
**Caloric Restriction Mimetics**	Rapamycin	0.5 µg/mL	N2A and SH-SYSY cells TDP-25	24 h	↑flux			[175]
		I.P. injection 2.24 mg/kg/day	SOD1-G93A mice	Long-term (pre-symptomatic)			↔	[174]
		I.P. injection 2.24 mg/kg/day	SOD1-H46R/H48Q mice	Long-term (pre-symptomatic)			↔	[174]
		I.P. injection 10 mg/kg 3×/week	FTLD-U mice (M)	Short-term (symptomatic)	↑flux	↑function		[176]
		I.P. injected 2 mg/kg/day	SOD1-G93A mice	Long-term (pre-symptomatic)	↑induction ↓degradation	↓number	↓	[177]
		Oral 2.33 mg/kg/day	SOD1-G93A mice	Long-term (pre-symptomatic)	↑induction		↔	[178]
		Oral 2.33 mg/kg/day	RAG1(-/-) x SOD1-G93A mice	Long-term (pre-symptomatic)	↑degradation		↑	[178]
	Trehalose	I.P. injection 2 g/kg 3×/week	SOD1-G86R mice	Long-term (pre-symptomatic)	↑flux	↑function	↑	[179]
		100 mM	NSC-34 cells SOD1-G86R	24 h	↑flux			[179]
		Oral 2% *w/v*/day	SOD1-G93A mice (F)	Short-term (pre-symptomatic)	↑flux	↑function		[180]
		Oral 2% *w/v*/day	SOD1-G93A mice (F)	Long-term (pre-symptomatic)	↑induction		↔	[180]
		100 mM	NSC-34 cells SOD1-G93A	24 h	↑flux			[180]
		Oral 2% *w/v*/day	SOD1-G93A mice (M)	Long-term (pre-symptomatic)	↑degradation	↑number	↑	[181]
		10 mM	NSC-34 cells SOD1-G93A	72 h	↑degradation			[182]
	Spermidine	I.P. injection 50 mg/kg 3×/week	FTLD-U mice (M)	Short-term (symptomatic)	↑flux	↑function ↑number		[176]
	Resveratrol	Oral 160 mg/kg/day	SOD1-G93A mice	Short-term (pre-symptomatic and symptomatic)	↑induction	↑function	↑	[183]
		I.P. injection 20 mg/kg 2×/week	SOD1-G93A mice (M)	Long-term (pre-symptomatic)		↑function	↑	[184]
		Oral 25 mg/kg/day	SOD1-G93A mice (F)	Long-term (pre-symptomatic)		↔		[185]
		10 µM	VSC4.1 cells SOD1-G93A	24 h				[186]
		I.P. injection 25 mg/kg/day	SOD1-G93A mice (M)	Long-term (pre-symptomatic)		↑number	↑	[187]
	Metformin	Oral 2 mg/mL	SOD1-G93A mice	Short-term (pre-symptomatic)		↑number	↔ (M) ↓ (F)	[188]
**Hormone Therapy**	Raloxifene	0.1 µM	NSC-34 cells TDP-25	24 h	↑degradation			[189]
	Tamoxifen	S.C. injection 50 mg/kg 3×/week	FTLD-U mice (M)	Short-term (symptomatic)	↑flux	↑function ↑number		[176]
	Progesterone	I.P. injection 4 mg/kg/day	SOD1-G93A mice (M)	Long-term (pre-symptomatic)	↑degradation		↑	[190]
**Cancer Therapy**	Bosutinib	10 µM	Patient iPSC derived MNs SOD1-L144FVX, SOD1-G93S, TDP-43 M337V, TDP-43 Q343R, TDP-43 G298S, C9orf72, SALS	7 days	↑degradation	↑number		[191]
		I.P. injection 5 mg/kg/day	SOD1-G93A mice	Short-term	↑degradation	↑number	↑	[191]
	Dasatinib	Oral gavage 25 mg/kg/day	SOD1-G93A mice	Long-term (pre-symptomatic)		↑function	↑	[192]
		Oral gavage 5 mg/kg/day	SOD1-G93A mice	Long-term (pre-symptomatic)			↔	[192]
**Psychotropic Therapy**	Lithium	I.P. injection 1 mEq/kg/day	SOD1-G93A mice (M)	Long-term (symptomatic)	↑flux	↑function	↑	[193]
		Plasma range of 0.4–0.8 mmol/L	Patients	15 months			↑	[193]
		Plasma range of 0.4–0.8 mmol/L	Patients	18 months			↔	[194]
		Plasma range of 0.4–0.8 mmol/L	Patients	12 months			↔	[195]
		Lithium carbonate 200 mg/kg/day OR Neu2000 30 mg/kg/day OR combination	SOD1-G93A mice	Long-term (pre-symptomatic)		↑function ↑number ↑↑co	↑ ↑↑co	[196]
		Plasma range of 0.4–0.8 mmol/L	Patients	15 months			↔	[197]
		Plasma range of 0.4–0.8 mmol/L	Patients	16 months			↔	[198]
		Plasma range of 0.3–0.8 mmol/L	Patients	13 months			↔	[199]
	Valproate or valproic acid	Oral 0.26% *w/v* 530 mg/kg/day	SOD1-G93A mice (M)	Long-term (pre-symptomatic)			↑	[200]
		1–2 mM	SH-SY5Y cells TDP-25	48 h	↑ induction			[201]
		I.P. injection 250 mg/kg/day	SOD1-G86R mice	Long-term (pre-symptomatic)		↑number	↔	[202]
		1500 mg	Patients	12 months			↔	[203]
	Valproic acid + lithium carbonate	0.3–0.75 mmol/L	Patients	18 months			↑	[204]
		I.P. injection lithium OR R valproate OR combination 60 mg/kg/2× day	SOD1-G93A mice	Long-term (pre-symptomatic)		↑function ↑↑co	↑ ↑↑co	[205]
	Carbamazepine	50 mg/kg 3×/week	FTLD-U mice (M)	Short-term (symptomatic)	↑flux	↑function ↑number		[176]
	Fluphenazine	5 µM	Primary neurons TDP43-A315T	48 h	↑flux		↑	[206]
	Methotrimeprazine	5 µM	Primary neurons TDP43-A315T	48 h	↑flux		↑	[206]
	Pimozide	I.P. injection 1 mg/kg/every 2 days	SOD1-G93A mice	Long-term (pre-symptomatic)	↓degradation	↓function ↔number	↓	[207]
		I.P. injection 1 mg/kg/every 2 days	SOD1-G93A mice	Long-term (symptomatic)	↔degradation	↓function ↔number	↓	[207]
		I.P. injection 1 mg/kg/every 2 days	TDP43-A315T mice	Short-term	↓degradation	↓function	↓	[207]
		1 mg/day	Patients	3–12 months			↑	[208]
**Anti- hypertensive**	Verapamil	240 mg/day	Patients	6 months			↔	[209]
	Rilmenidine	I.P. injection 10 mg/kg 4×/week	SOD1-G93A mice	Long-term (pre-symptomatic)	↑induction	↓number	↓	[210]
		10 µM	NSC-34 cells SOD1- A4V	24 h	↑degradation *			[210]
		10 µM	Stem cells SOD1-WT	24 h	↑flux			[210]
	Berberine	10–30 µg/ml	N2a cells TDP-25	6 h, 24 h	↑flux			[211]
**Anti-histamine**	Latrepiradine	I.P. injection I µg/kg/day	SOD1-G93A mice	Short-term (pre-symptomatic)	↑induction	↑function (M)	↑ (M)	[212]
	Clemastine	I.P. injection 10 mg/kg/5×/week	SOD1- G93A mice (F)	Long-term (pre-symptomatic)		↑number	↔	[213]
		I.P. injection 50 mg/kg/5×/week	SOD1-G93A mice (F)	Short-term (pre-symptomatic)	↑flux	↑function	↑	[214]
		I.P. injection 50 mg/kg 5×/week	SOD1-G93A mice (F)	Long-term (pre-symptomatic)		↔	↔	[214]
		30 µM	NSC-34 cells SOD1-G93A	6 h, 24 h	↑flux (6 hrs) ↓flux (24 hrs)			[214]

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
