# Peer review of "Amyotrophic Lateral Sclerosis and Autophagy: Dysfunction and Therapeutic Targeting"

_cells, 2020, doi:10.3390/cells9112413_

Round 1

Reviewer 1 Report

This review article is well organized, clear, and thorough, with clear and well-designed figures and tables

The authors need to standardize the font and maybe check the literature

Reviewer 2 Report

Manuscript ID: cells-972283

In this review entitled ‘Amyotrophic lateral sclerosis and autophagy: dysfunction and therapeutic targeting’ Amin et al. present the state of the art of the pathobiology of ALS with focus on autophagy. Authors present critical overview of various therapeutic targeting schemes applied recently and undergoing trials.   

General comments

This is a timely review. It is well structured and generally well written. It presents a comprehensive analysis of the topic. The manuscript presents the state of the art of ALS, followed by the biology of protein misfolding and neuro-degradation, moving onto autophagy and therapeutic advances. It discusses principal areas of recent developments in therapies of autophagy. The quality of illustrations is fair, conclusions are sound.

Specific notes and recommendations

The manuscript will benefit from (1) stating clearly its novelty (2) inclusion in the discussion of most recent studies in the area (i.e. from 2019-2020 there were 2 references, which is not acceptable) (3) highlighting clearly the outstanding problems and (3) stressing that ALS is a motor neuron disease, therefore, the retro-axonal delivery and therapeutic targeting must be considered. These must be discussed (at lease delivery methods and recent advances). In this context, the authors should update the references and discussions by following studies: (1) DOI: 10.1021/mp400247t; DOI: 10.1007/s00429-015-1004-0; DOI: 10.1016/j.molmed.2016.09.004; DOI: 10.1007/s00429-015-1004-0

The final recommendation will be made based on the quality of the revision of the manuscript. 

Reviewer 3 Report

The review describes autophagy and the proposed role of autophagy in ALS via the development of toxic intracellular protein aggregates.  The review summarises known ALS genes associated with autophagy and includes a comprehensive consideration of autophagy manipulation in ALS disease models.  I enjoyed reading this review and I found it informative.

Minor points:

- Heritability of sporadic ALS is estimated to be ~52% -- this should be corrected in the manuscript.

- Please mention the observation that neuronal loss is correlated with the quantity of TDP-43 pathology.  This is not true for any other ALS-associated molecular pathology and is key evidence for an upstream role of pathology in neurodegeneration.

Round 2

Reviewer 2 Report

I recommend this work for publishing.